# Seed Coating with Thiamethoxam-Induced Plant Volatiles Mediates the Olfactory Behavior of *Sitobion miscanthi*

**DOI:** 10.3390/insects15100810

**Published:** 2024-10-16

**Authors:** Jiacong Sun, Yonggang Liu, Shaodan Fei, Yixuan Wang, Jinglong Liu, Haiying Zhang

**Affiliations:** 1College of Plant Protection, Gansu Agricultural University, Lanzhou 730070, China; sunjiacong2023@163.com (J.S.); wangyixuan624@163.com (Y.W.); liujinglong1999@163.com (J.L.); 2Institute of Plant Protection, Gansu Academy of Agricultural Sciences, Lanzhou 730070, China; feishaodan@163.com

**Keywords:** *Sitobion miscanthi*, thiamethoxam, olfactory behavior, wheat, insect–plant interaction

## Abstract

**Simple Summary:**

This work investigates the olfactory mechanism of aphids with varying preferences for TMX-coated and uncoated wheat plants. This study utilized the olfactory localization preferences of aphids to identify the volatile metabolites of both TMX-coated and uncoated wheat. The differential metabolites were then used to assess the olfactory behavior of aphids. The results indicated that four compounds had repellent activity towards aphids, while two compounds demonstrated attracting activity towards aphids. These findings highlight the potential of using TMX-induced VOCs to manage aphid behavior and develop environmentally friendly pest control strategies.

**Abstract:**

Pesticides can induce target plants to release odors that are attractive or repellent to their herbivore insects. But, to date, the activity of volatile organic compounds (VOCs), singly or as mixtures, which play a crucial role in the olfactory behavior of herbivore insects, remains unclear. The objective of our research was to investigate the impact of thiamethoxam (TMX), a pesticide, on the emission of odors by wheat plants, and how these odors influence the behavior of grain aphids (*Sitobion miscanthi*). *S. miscanthi* showed a greater repellent response to the volatiles emitted by Thx-induced plants compared to those emitted by uncoated plants. Using gas chromatography–mass spectrometry (GCMS), we discovered that TMX greatly induced the release of VOCs in wheat plants. For instance, the levels of Bornyl acetate, 2-Oxepanone, Methyl acrylate, Cyclohexene, α-Pinene, and 1-Nonanol in coated wheat plants were significantly higher as compared to uncoated wheat plants. Moreover, varying concentrations also had an impact on the olfactory behavior of *S. miscanthi.* For instance, Cyclohexene exhibited clear attractiveness to aphids at concentrations of 100 μL/mL, whereas it displayed evident repellent properties at concentrations of 1 μL/mL and 10 μL/mL. These new findings demonstrate how TMX-induced VOCs affect the behavior of *S. miscanthi* and could help in developing innovative approaches to manage aphids by manipulating the emission of plant volatiles. Furthermore, these findings can also be utilized to evaluate substances that either attract or repel aphids, with the aim of implementing early monitoring and environmentally friendly methods to manage aphids, while simultaneously impeding the spread of viruses.

## 1. Introduction

Wheat is the second most extensively cultivated crop globally, with an estimated area of 200 million hectares [1]. Wheat is utilized worldwide to produce bread, pasta, and other bakery items, as well as to a small extent for industrial products [2]. Nevertheless, wheat is highly susceptible to aphid infestations, particularly by *Sitobion miscanthi*, which is considered one of the most economically significant insect pests [3]. *S. miscanthi* is a globally distributed sap-sucking specialist of cereal and a dominant species in wheat-growing regions across China [4]. It threatens wheat production in various ways such as pillaging nutrition from the host, transmitting pathogenic plant viruses, and defecating sticky honeydew that further obstructs photosynthesis and reduces wheat quality [5]. It is now one of the dominant species in the middle and late stages of wheat growth in China [6]. Insects typically perceive and distinguish odors by olfaction, which leads to various actions such as evading predators, laying eggs, seeking mates, and foraging for food [7]. Insects have the ability to detect the chemical compounds produced by both host and non-host plants using their sense of smell. These compounds, known as volatile metabolites, can either repel or attract insects, making them useful as repellents or attractants [8]. Currently, the primary method for controlling wheat aphids is by the use of chemical pesticides, specifically neonicotinoid insecticides [9,10]. Syngenta has produced thiamethoxam, a second-generation neonicotinoid that possesses a distinctive structure and displays exceptional insecticidal properties [11]. Due to the incorporation of the cloprothiazole group into its molecular structure, it has significant efficacy against pests with piercing–sucking and chewing mouthparts [12]. It is extensively utilized in diverse planting methods [13] and exerts advantageous impacts on the growth and development of plants [14]. Nevertheless, the utilization of insecticides has resulted in other issues, including diminished control effectiveness and heightened aphid resistance [15]. As an illustration, 9% of the *Myzus persicae* population in Greece has shown significant resistance (24–73 times higher) to imidacloprid [16]. *Aphis gossypii* Glover in China has exhibited a significant degree of resistance to imidacloprid, with resistance ratios over 1200-fold [17]. The formation of insecticide resistance is a dynamic evolutionary result of long-term, widespread, and extensive use of insecticides [18]. Therefore, it is necessary to reduce the use of chemical pesticides and adopt environmentally friendly methods to control aphids.

Plants naturally produce and emit a diverse range of volatile organic chemicals, which play a vital role in plant reproduction and defense [19]. Plant volatile chemicals serve as a means of communication and interaction between plants and their surrounding environment [20]. So far, researchers have identified 1700 volatile chemicals from over 90 plant families. Plants exude these volatile compounds from their leaves, flowers, and fruits into the air, and from their roots into the soil, to protect themselves against herbivores and pathogens [21]. Plant volatile chemicals encompass a variety of substances such as isoprene, terpene, fatty acid derivatives, alcohols, esters, volatile oils, plant development regulators (abscisic acid, auxin, cytokinin, etc.), phenolic compounds, and secondary metabolites [22]. Consequently, in recent times, scientists have used volatile organic molecules as a means to shield plants from both biological and abiotic challenges, as well as interactions with their surroundings [23].

Currently, there has been limited focus on the study of how pesticides affect the release of plant volatiles and how this impacts the olfactory behavior of aphids towards their host plants. Thus, the objectives of the study were to (i) evaluate behavioral responses of *S. miscanthi* to volatile compounds released from TMX-coated and uncoated wheat plants using an olfactometer, (ii) identify and quantify VOCs emitted by TMX-coated and uncoated wheat plants by gas chromatography–mass spectrometry (GC-MS), (iii) analyze the variations in the composition of VOCs in TMX-coated and untreated wheat plants, and (iv) assess the role of individual VOCs to observe whether these volatile compounds could act as olfactory cues to attract or repel *S. miscanthi*. Our investigation revealed that aphids exhibit distinct smell preferences towards TMX-coated and untreated wheat plants. Thus, we propose that distinct olfactory reactions are triggered by diverse volatile metabolites. Additionally, we examined the olfactory preference of aphids for specific chemicals. This research has the potential to aid in the development of attractants and repellents for managing wheat aphids, without the need for chemical insecticides.

## 2. Materials and Methods

### 2.1. Samples

Aphids (*Sitobion miscanthi*) were initially obtained from fields located in Lanzhou City, Gansu Province, China (36.100324° N, 103.688190° E). They were introduced into an artificial climate incubator with specific temperature conditions, (18 ± 1) °C at night and (20 ± 1) °C during the day, along with a relative humidity of 60–70% and a photoperiod of 16 h of light followed by 8 h of darkness. These aphids were not exposed to any pesticides and were provided with wheat leaves as their food source for several generations.

Commercially sourced wheat seeds (*Triticum aestivum* L., Longchun 30) were acquired. The seeds were cultivated in plastic pots with dimensions of 10 cm in height and 4 cm in diameter. The cultivation took place in an incubator with controlled conditions, maintaining a temperature of 25 ± 3 °C, relative humidity of 55 ± 10%, and a light–dark cycle of 16 h of light followed by 8 h of darkness. No insecticides were applied throughout the cultivation process.

### 2.2. Aphid Preference for TMX-Coated Versus Uncoated Wheat Plants

The wheat seeds, coated with TMX, were planted in a nutrient bowl (Heng De li Garden Supplies Co., Ltd., Guangzhou, China) that had a diameter of 15 cm and a height of 8 cm. Once the wheat reached a height of around 13 cm, it was selected for the experiment, while the uncoated wheat was employed as the control.

Wheat plants were enclosed within bug cages of 1.8 m in length, 0.6 m in width, and 0.5 m in height. Within each rearing cage, there were wheat plants infested with 30 adult aphids positioned at the center. Additionally, there were a pot of coated wheat and a pot of uncoated wheat put at a distance of 10 cm from the center in opposite directions. A count of the aphid population on the plants was documented at 24 h intervals for a total of 10 repetitions [24].

A total of ten aphids were put into a four-arm olfactometer (Xuelai Biotechnology Co., Ltd., Nanjing, China) for each trial. The aphids were then examined for a duration of 10 min. To avoid any potential spatial variations, the olfactometer was rotated by 90 degrees for each trial. The quantities of aphids that entered the flavor source arm, the control arm, and remained in the activity room were documented, with each treatment being repeated 10 times.

The olfactory response of aphids to TMX-coated wheat leaves was quantified using a four-arm olfactometer. Five wheat strains coated with thiamethoxam were placed in two opposite flavor source bottles (Xuelai Biotechnology Co., Ltd., Nanjing, China) (100 mL) as the treatment condition. Five uncoated wheat plants were placed in the other two taste source bottles (100 mL) as the control condition. The height of the wheat plants was around 13 cm.

The olfactometer comprises a center circular glass control chamber with a diameter of 5 cm and four arms with an inner diameter of 1 cm and a length of 7 cm. Each arm is equipped with a glass elbow measuring 5 cm in length and a glass pear-shaped bottle attached in an upward direction. The filtered and humidified air is introduced into each air source bottle via a Teflon tube at a rate of 0.15 L/min (controlled by a manifold equipped with four flow meters). This air then transports the volatile organic compounds to the olfactometer’s testing region. After each test, the olfactometer was cleansed with distilled water to prevent the buildup of chemicals from the host or insects. The experiments were performed at a constant room temperature of 20 ± 2 °C and a relative humidity of 55 ± 5% [25]. The data collected from the experiment were organized using Excel 2021, and the inclination of aphids towards wheat plants was examined using a *t*-test in SPSS Statistics 26 software.

### 2.3. Aphids’ Olfactory Behavioral Reactions to Individual Chemicals

The individual component to be analyzed was dissolved in n-hexane company, city, and country, and five concentration gradients (100 μL/mL, 10 μL/mL, 1 μL/mL, 0.1 μL/mL, 0.01 μL/mL) were established. A volume of 20 microliters was added to a filter paper measuring 1 square centimeter. The filter paper was then placed in two flavor source bottles, one on each side, as a treatment. The other two flavor bottles were loaded with 20 microliters of n-hexane as a control [26].

Each trial of the experiment involved introducing ten aphids into an olfactometer, observing them for a duration of ten minutes, and rotating the olfactometer by 90 degrees to ensure that any spatial variations were eliminated. The quantities of aphids that entered the source arm, control arm, and remained in the activity room were documented accordingly. The experiment involved conducting each treatment 10 times and replacing the filter paper after each repetition [27] (Figure 1). The avoidance rate is determined using the following formula: Avoidance rate (Rate) = [(number of aphids in the source arm − number of aphids in the control arm)/(number of aphids in the source arm + number of aphids in the control arm)]. The avoidance rate is a measure of the aphids’ positive or negative taxis towards flavor source material. A value greater than 0 indicates positive taxis, with a larger value suggesting stronger taxis (up to a maximum of 1). Conversely, a value less than 0 indicates negative taxis towards the flavor source material.

### 2.4. Analysis of Volatile Chemicals in Wheat Plants Treated with TMX-Coated and Untreated Plants

#### 2.4.1. Sample Preparation and Treatment

The materials were severed, measured, promptly frozen in liquid nitrogen, and preserved at −80 °C until required. Samples were ground to a powder in liquid nitrogen. The samples were dispatched to NoveBio Company, located in Xi’an, China, for testing. A total of 500 mg (1 mL) of the powder was transferred immediately to a 20 mL headspace vial (Agilent, Palo Alto, CA, USA), containing NaCl saturated solution, to inhibit any enzyme reaction. The vials were sealed using crimp-top caps with TFE-silicone headspace septa (Agilent). At the time of SPME analysis, each vial was placed in 60 °C for 5 min, then a 120 µm DVB/CWR/PDMS fiber (Agilent, Palo Alto, CA, USA) was exposed to the headspace of the sample for 15 min at 60 °C.

#### 2.4.2. Gas Chromatography–Mass Spectrometry Conditions

After sampling, desorption of the VOCs from the fiber coating was carried out in the injection port of the GC apparatus (Model 8890; Agilent, Palo Alto, CA, USA) at 250 °C for 5 min in the splitless mode. The identification and quantification of VOCs was carried out using an Agilent Model 8890 GC and a 7000D mass spectrometer (Agilent) equipped with a 30 m × 0.25 mm × 0.25 μm DB-5MS (5% phenyl-polymethylsiloxane) capillary column. Helium was used as the carrier gas at a linear velocity of 1.2 mL/min. The injector temperature was kept at 250 °C and the detector was kept at 280 °C. The oven temperature was programmed from 40 °C (3.5 min), increasing at 10 °C/min to 100 °C, at 7 °C/min to 180 °C, at 25 °C/min to 280 °C, and then holding for 5 min. Mass spectra were recorded in electron impact (EI) ionization mode at 70 eV. The quadrupole mass detector, ion source, and transfer line temperatures were set, respectively, at 150, 230, and 280 °C. The MS was in selected ion monitoring (SIM) mode and was used for the identification and quantification of analytes. The mixed samples were performed per three samples and the total ion flow diagrams of mixed samples were used to judge the reproducibility of the results. Identification of volatile compounds was achieved by comparing the mass spectra with the data system library (MWGC or National Institute of Standards and Technology) and linear retention index.

#### 2.4.3. Data Analysis

The metabolomics data were normalized using R version 4.3.0 (21 April 2023). MetaboAnalyst 4.0 (15 July 2019) was used to analyze the metabolites using the orthogonal partial least squares (OPLS) model. R2X (the interpretability of the model for the categorical variable X), R2Y (the interpretability of the model for the categorical variable Y), and Q2 (predictability of the model) were obtained after cross-validation to judge the validity of the model.

Unsupervised PCA (principal component analysis) was performed by statistics function prcomp within R version 4.3.0 (21 April 2023). The data were unit variance scaled before unsupervised PCA. The HCA (hierarchical cluster analysis) results of samples and metabolites were presented as heatmaps with dendrograms.

#### 2.4.4. Differential Metabolites Selected

For two-group analysis, differential metabolites were determined by VIP (VIP ≥ 1) or absolute Log_2_FC (|Log_2_FC| ≥ 1.0). VIP values were extracted from OPLS-DA results, which also contain score plots and permutation plots, and were generated using R package MetaboAnalystR. The data were log(log_2_) transformed and underwent mean centering before OPLS-DA. To avoid overfitting, a permutation test (200 permutations) was performed.

#### 2.4.5. Analysis of *S. miscanthi*’s Olfactory Preference

The olfactory behavior selection of aphids towards different plants and individual chemicals was evaluated using a *t*-test in SPSS 26.

## 3. Results

### 3.1. Preference of S. miscanthi for TMX-Coated and Uncoated Wheat Plants

The olfactory preference of *S. miscanthi* for wheat plants with different treatments was documented for five consecutive days. On the first day (*p* = 0.074), aphids did not display any discernible olfactory preference between the two distinct wheat plants. From the second day onwards, the aphids showed a substantially higher preference for uncoated wheat plants compared to coated wheat plants (*p* = 0.037). This phenomenon remained virtually unaltered over the whole observation period of the wheat. By the fifth day, the disparity in aphid population density between TMX-coated and uncoated wheat plants reached its highest point (Figure 2).

The olfactory preference of aphids for different wheat plants was assessed using a four-arm olfactometer. The findings indicated that the scent emitted by the coated wheat had a deterrent impact on the majority of aphids, but the uncoated wheat had an enticing effect on the majority of aphids (Figure 3A, *p* = 0.001). The results suggest that wheat plants coated with TMX are more effective in repelling aphids compared to uncoated plants. As a result, most aphids show a preference for uncoated wheat plants in the olfactometer.

### 3.2. Analysis of Volatile Compounds Emitted by TMX-Coated and Untreated Wheat Plants

#### 3.2.1. OPLS-DA and PCA

To investigate the differences in the metabolic compositions between TMX-coated wheat plants and uncoated controls, we performed a metabolic analysis. The orthogonal partial least squares discriminant analysis (OPLS-DA) was used to discriminate metabolic profiles between the groups of TMX-coated wheat plants and uncoated controls (Figure 4 and Figure 5).

Principal component analysis revealed distinct spatial enrichment patterns of volatiles among several plants (Figure 6), showing considerable differences between them. Furthermore, it is possible to combine each set of replicates, suggesting that the chemical analysis data are dependable and suitable for further examination. PC1 accounted for 72.21% of the variation, whereas PC2 accounted for 15.91%. The volatile compounds of S1 and SK plants can be distinguished from PC1, suggesting variations in the volatile compounds of coated and uncoated wheat plants [28]. Furthermore, by combining FC with VIP from the OPLS-DA model, 34 metabolites were finally altered in the TMX-coated seedlings, which was considerably greater than those in uncoated plants (Table 1).

#### 3.2.2. Metabolite Hierarchical Clustering and Difference Significance Analysis

This article used hierarchical clustering and difference significance analysis of metabolites to identify potential compounds that either attract or repel aphids. Our objective was to identify compounds that have a positive or negative correlation with aphid olfactory preference by coating wheat seeds with TMX.

Initially, by hierarchical cluster analysis, a total of nine metabolite categories were gathered. Esters and terpenoids are the predominant metabolites, as seen in Figure 7. Furthermore, the alterations in each category of metabolites vary. For instance, certain esters exhibit a high concentration in coated plants, but others have a low concentration in coated plants. Furthermore, to investigate the metabolites associated with aphid selection preference, significant difference analysis revealed that the volatiles emitted by TMX-coated wheat plants were considerably increased compared to uncoated plants. Additionally, there were 34 distinct metabolites identified (Figure 8).

### 3.3. Quantification of Aphids’ Olfactory Response to Specific Chemicals

In order to assess the impact of volatile metabolites on aphid orientation behavior, the behavioral reactions of aphids to various potential chemicals were examined using a four-arm olfactometer. Table 2 displays the pertinent information for the six chemicals that were acquired from the market. Initially, the olfactory orientation behavior of aphids towards air was examined in an olfactometer. The results indicated that there was no statistically significant disparity in the preference of aphids for air in either arm (Figure 3B). This suggests that the apparatus has excellent airtightness and may be effectively utilized for conducting olfactory studies on standard chemicals with aphids.

#### 3.3.1. The Behavioral Reactions of *S. miscanthi* to Three Terpenoids

Borneol acetate demonstrated significant repellent effects on aphids at concentrations of 100 μL/mL (*p* < 0.01), 10 μL/mL (*p* < 0.01), 1 μL/mL (*p* < 0.01), and 0.1 μL/mL (*p* < 0.05) (Figure 3C). α-Pinene, at concentrations of 1 μL/mL and 10 μL/mL, exhibited significant attraction effects on aphids (*p* < 0.05), while other concentrations did not show noticeable attraction effects compared to the control group (Figure 3D). Cyclohexene displayed significant attraction effects on aphids at a concentration of 100 μL/mL (*p* < 0.05) but exhibited significant repellent effects at concentrations of 1 μL/mL and 10 μL/mL (*p* < 0.05) (Figure 3E). At a dose of 0.01 μL/mL, these three terpenoids showed no statistically significant difference in the number of aphids between the treatment group and the control group (*p* > 0.05). The findings indicate that nearly all levels of Borneol acetate demonstrated repellent effects on aphids. At moderate concentrations (1 μL/mL and 10 μL/mL), α-pinene exhibited attractive properties towards aphids. Cyclohexene, at high concentrations (100 μL/mL), displayed attractive properties towards aphids, while at moderate concentrations (1 μL/mL and 10 μL/mL), it exhibited repellent properties towards aphids.

#### 3.3.2. The Behavioral Reactions of *S. miscanthi* to Two Esters

In the tested ester compounds (Figure 3F,G), 2-Oxepanone and Methyl acrylate exhibited strong aversive effects on aphids at a concentration of 100 μL/mL (*p* < 0.05). However, at other concentrations (10 μL/mL, 1 μL/mL, 0.1 μL/mL, 0.01 μL/mL), there was no significant disparity in the number of aphids between the treated and control groups (*p* > 0.05).

#### 3.3.3. The Behavioral Reactions of *S. miscanthi* to Alcohols

At a concentration of 1 μL/mL (*p* < 0.05), 1-Nonanol exhibited a notable tendency to repel aphids (*p* < 0.05). However, at other doses, there was no significant avoidance observed when compared to the control (Figure 3H).

## 4. Discussion

This work investigates the olfactory mechanism of aphids with varying preferences for TMX-coated and uncoated wheat plants, as there is currently limited research on the subject. The bioassay of aphid olfactory behavior demonstrated that TMX-coated wheat plants had a substantial deterrent impact on aphids, as compared to the control group of wheat (Figure 3A). This could be attributed to the modification of the volatile composition in wheat plants caused by pesticides, which in turn affects the ability of aphids to detect and locate odors. Our findings corroborate prior research indicating that plant defense mechanisms, including alterations in volatile organic compound emissions, are frequently activated by chemical treatments [28]. Aphids depend significantly on their olfactory receptors to perceive and react to plant volatiles. Recent research indicates that aphid olfactory systems are highly sensitive to specific compounds, such as terpenoids, which can trigger either attraction or repellence [29]. This study utilized the olfactory localization preference of aphids to identify the volatile metabolites of both TMX-coated and uncoated wheat. The differential metabolites were then used to assess the olfactory behavior of aphids. The results indicated that four compounds had repellent activity towards aphids, while two compounds demonstrated attracting activity towards aphids. These compounds were expected to be developed as repellents or attractants for the aim of controlling aphids. The metabolomics analysis revealed notable disparities in volatile components between wheat plants coated with TMX and those that were not coated, as depicted in Figure 6. TMX-coated wheat plants have a comparatively high concentration of terpenoids, particularly Borneol acetate and caprolactone. Prior research has shown that terpenoids and other volatile organic compounds, including green leaf volatiles, significantly influence insect behavior, making these substances essential targets for pest management [30]. The findings also indicated notable variations in the impact of the identical substance on the olfactory behavior of *S. miscanthi* at varying doses. Thus, this work suggests that the olfactory system of aphids may resemble human hearing, which operates within a frequency range of 20 Hz to 20,000 Hz [31]. Insects, particularly *S. miscanthi*, will only exhibit olfactory behavioral reactions within a specific concentration range of volatiles. If the concentration of volatiles exceeds this range, the insects will encounter olfactory behavioral failure. Furthermore, this paper discovered that some chemicals, such Cylohexene, display appealing properties to aphids when present in high concentrations (100 μL/mL). However, when present in low quantities (1 μL/mL, 10 μL/mL), these compounds demonstrate repellent characteristics towards aphids. This finding was also corroborated in the study conducted by Guo Xiangling et al. [32]. The results demonstrated that there exists a threshold for the dose response of aphids to the chemicals. A notable response can only be triggered when the threshold is surpassed, while a large dose would have a contrary impact.

This article presents information on eco-friendly ways for controlling aphids, as well as valuable insights into the aphid olfactory system, which can aid in the creation of repellents and attractants. Previous studies have also shown that these volatile signals can influence the behavior of aphids’ natural enemies, such as parasitoids and predators, which are attracted to herbivore-induced plant volatiles [33]. Incorporating these findings into the push–pull strategy, originally described by Pickett et al. and successfully applied in sub-Saharan Africa, could significantly reduce pesticide usage while enhancing biological control efforts [34].

In conclusion, our study offers significant insights into the influence of VOCs on aphid behavior, which may facilitate the creation of attractants and repellents. Furthermore, this research supports the push–pull model, where plant volatiles play an integral role in pest management. The application of these findings may also mitigate the spread of viral diseases, often transmitted by aphids, offering a dual benefit for agricultural sustainability.

## Figures and Tables

**Figure 1 insects-15-00810-f001:**
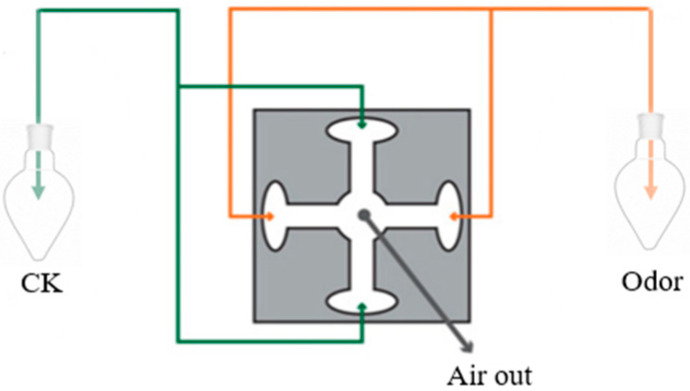
Aphids’ olfactory behavioral reactions to individual chemicals.

**Figure 2 insects-15-00810-f002:**
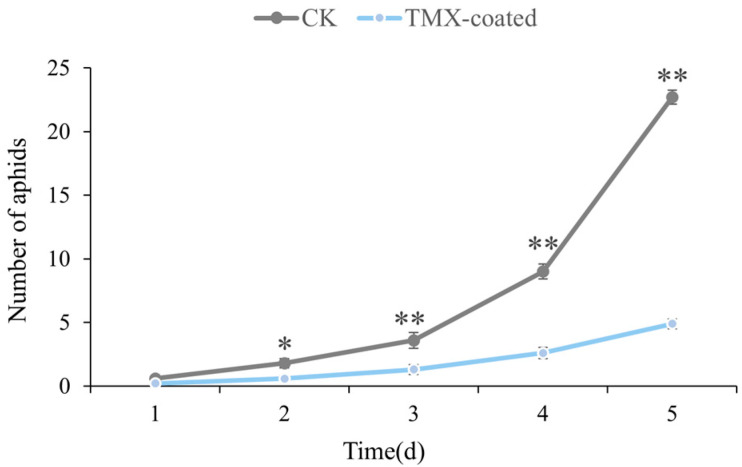
*S. miscanthi*’s olfactory preference for TMX-coated and uncoated potted wheat within 5 consecutive days. (* *p* < 0.05, ** *p* < 0.01).

**Figure 3 insects-15-00810-f003:**
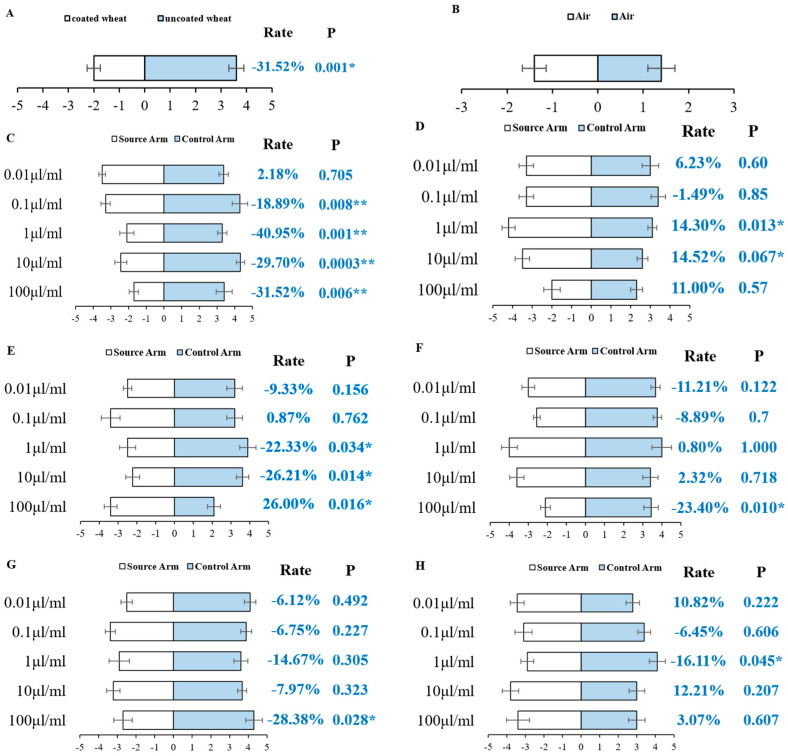
Quantification of aphids’ olfactory response to wheat plants and specific chemicals. The avoidance rate (Rate) is a measure of the aphids’ positive or negative taxis towards flavor source material. A value greater than 0 indicates positive taxis, with a larger value suggesting stronger taxis. Conversely, a value less than 0 indicates negative taxis towards the flavor source material. Asterisks indicate a significant difference (* *p* < 0.05, ** *p* < 0.01). 0–5 represents the average number of aphids on the source arm, and 0–(−5) represents the average number of aphids on the control arm. (**A**) Aphids’ olfactory preference for various wheat plants. (**B**) Aphids’ olfactory orientation behavior toward air. (**C**) Aphids’ behavioral responses to Borneol acetate. (**D**) Aphids’ behavioral responses to α-Pinene. (**E**) Aphids’ behavioral responses to Cyclohexene. (**F**) Aphids’ behavioral responses to 2-Oxepanone. (**G**) Aphids’ behavioral responses to Methyl acrylate. (**H**) Aphids’ behavioral responses to 1-Nonanol.

**Figure 4 insects-15-00810-f004:**
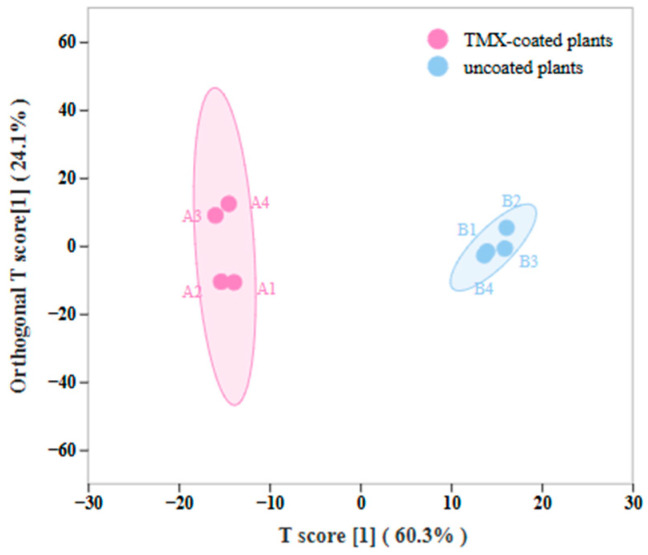
OPLS-DA showed the possible discrimination of metabolites from TMX-coated wheat plants (*n* = 4) and uncoated controls (*n* = 4) as indicated.

**Figure 5 insects-15-00810-f005:**
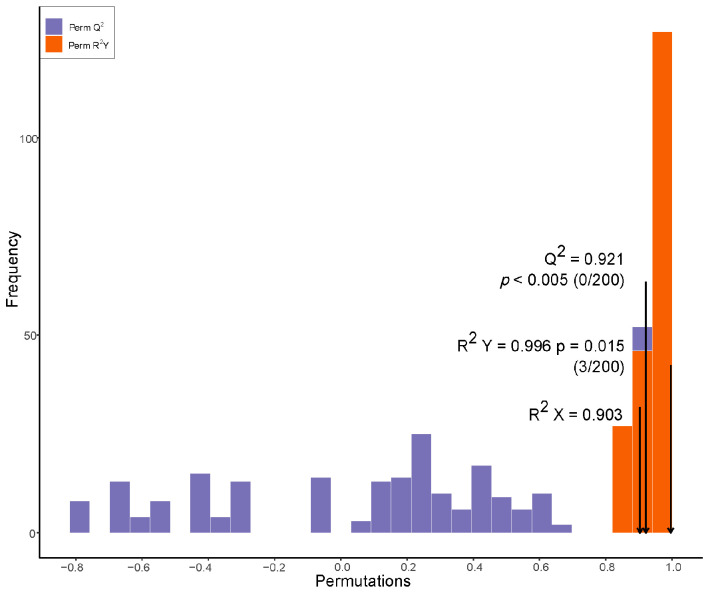
R^2^ and Q^2^ represent goodness of fit and prediction, respectively, and *p*-value shows the significance level of the model (x axis = predictive components, y axis = orthogonal component).

**Figure 6 insects-15-00810-f006:**
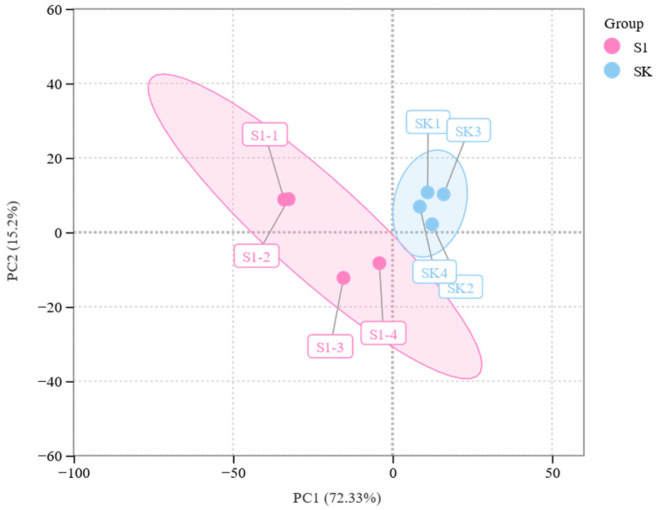
Principal component analysis. Each point represents a sample, and different colors represent different groups. The distance between the two groups represents the variance between them. S1: TMX-coated wheat; SK: uncoated wheat.

**Figure 7 insects-15-00810-f007:**
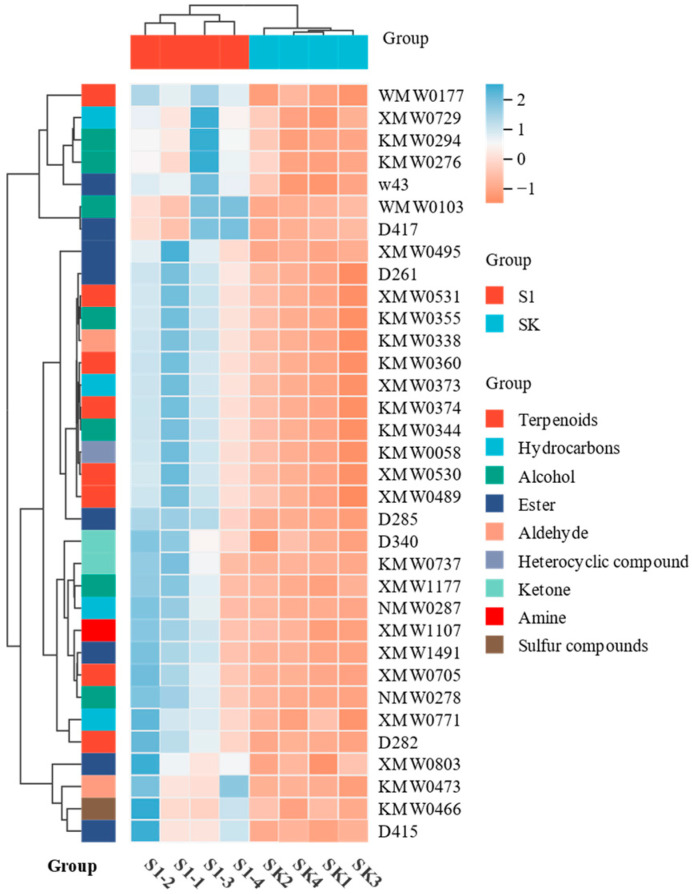
Metabolite hierarchical clustering of different wheats. S1: TMX-coated wheat; SK: uncoated wheat.

**Figure 8 insects-15-00810-f008:**
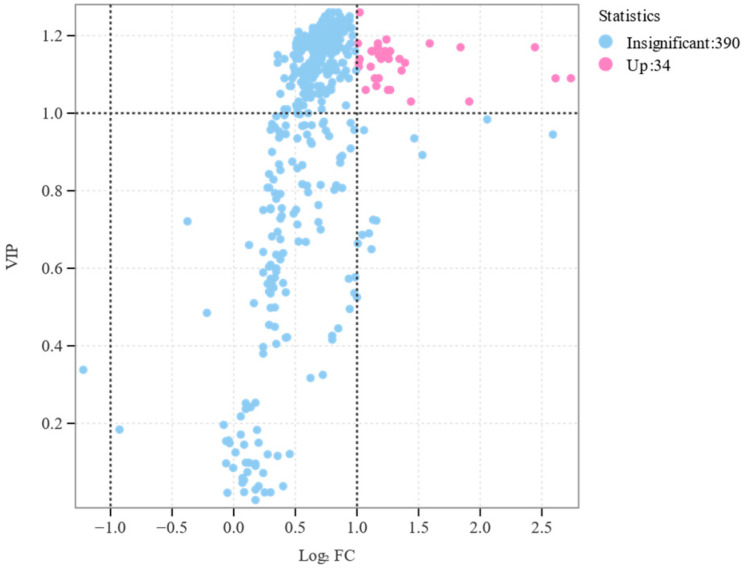
Difference significance analysis of different plants.

**Table 1 insects-15-00810-t001:** Differential volatile organic compounds (VOCs) emitted by TMX-coated and uncoated wheat plants.

Compounds	Formula	CAS	TMX-Coated	Uncoated	VIP	FC	Retention Index
Bornyl acetate	C_12_H_20_O_2_	76-49-3	2.83 × 10^3^	9.45	1.24	2.99 × 10^2^	1285
2-Oxepanone	C_6_H_10_O_2_	502-44-3	7.18 × 10^4^	3.58 × 10^4^	1.18	2.01	1065
Cyclohexene	C_10_H_16_	586-62-9	5.18 × 10^3^	2.42 × 10^3^	1.11	2.15	1088
Methyl acrylate	C_6_H_10_O_2_	924-50-5	1.12 × 10^5^	2.40 × 10^5^	1.66	4.65 × 10^−1^	842
α-Pinene	C_10_H_16_	80-56-8	1.99 × 10^3^	9.35	1.11	2.13 × 10^2^	937
1-Nonanol	C_9_H_20_O	143-08-8	4.88 × 10^5^	2.09 × 10^5^	1.16	2.34	1173
2-Isobutyryloxy-toluol	C_11_H_14_O_2_	36438-54-7	1.81 × 10^1^	3.33	1.17	5.44	1276
1,5-Cyclooctadiene,3,4-dimethyl	C_10_H_16_	21284-05-9	4.30 × 10^3^	2.12 × 10^3^	1.13	2.02	1046
α-Eudesmol	C_1_5H_26_O	473-16-5	3.38 × 10^3^	1.29 × 10^3^	1.13	2.62	1653
linalool oxide	C_10_H_18_O_2_	14049-11-7	3.70 × 10^5^	1.55 × 10^5^	1.15	2.38	1173
Iso-3-thujyl acetate	C_12_H_20_O_2_	62181-90-2	4.40 × 10^1^	1.17 × 10^1^	1.03	3.76	1298
Lavandulyl caproate	C_16_H_28_O_2_	59550-36-6	1.27 × 10^4^	4.94 × 10^3^	1.11	2.57	1650
Dehydromevalonic lactone	C_6_H_8_O_2_	2381-87-5	1.35 × 10^4^	4.48 × 10^3^	1.18	3.01	1169
a-Terpinen-7-al	C_10_H_14_O	1197-15-5	6.34 × 10^3^	1.77 × 10^3^	1.17	3.58	1283
2-Nonenal	C_9_H_16_O	18829-56-6	1.20 × 10^5^	5.52 × 10^4^	1.16	2.17	1162
3-Hexanone, 1-phenyl	C_12_H_16_O	29898-25-7	1.74 × 10^4^	8.57 × 10^3^	1.14	2.03	1427
trans,cis-2,6-Nonadien-1-ol	C_9_H_16_O	28069-72-9	3.03 × 10^5^	1.39 × 10^5^	1.16	2.18	1170
1-p-menthene-8-thiol	C_10_H_18_S	71159-90-5	8.38 × 10^1^	3.09 × 10^1^	1.03	2.71	1283
cis-3-Pinanone	C_10_H_16_O	15358-88-0	3.26 × 10^5^	1.44 × 10^5^	1.15	2.26	1173
p-Mentha-1,5-dien-8-ol	C_10_H_16_O	1686-20-0	7.93 × 10^5^	3.31 × 10^5^	1.14	2.39	1167
Undecane, 2-methyl	C_12_H_26_	7045-71-8	4.00 × 10^5^	1.66 × 10^5^	1.16	2.41	1164
Levomenthol	C_10_H_20_O	2216-51-5	5.71 × 10^5^	2.49 × 10^5^	1.14	2.29	1171
sec-butyl thioisovalerate	C_9_H_18_OS	2432-91-9	8.22 × 10^5^	3.66 × 10^5^	1.17	2.25	1174
2-Cyclopentylethanol	C_7_H_14_O	766-00-7	3.13 × 10^5^	5.12 × 10^4^	1.09	6.11	1003
3-Hexen-1-ol, acetate	C_8_H_14_O_2_	3681-82-1	2.00 × 10^5^	3.00 × 10^4^	1.09	6.66	1005
1-Octanol	C_8_H_18_O	111-87-5	1.32 × 10^5^	6.29 × 10^4^	1.06	2.10	1072
2-Octen-1-ol	C_8_H_16_O	18409-17-1	5.94 × 10^4^	2.94 × 10^4^	1.12	2.02	1067
p-Mentha-1(7),2-dien-8-ol	C_10_H_16_O	65293-09-6	8.84 × 10^5^	4.10 × 10^5^	1.12	2.16	1163
Methyl dihydrojasmonate	C_13_H_22_O_3_	24851-98-7	3.31 × 10^4^	1.41 × 10^4^	1.19	2.36	1656
*N*,*N*’-Methylenebismethacrylamide	C_9_H_14_N_2_O_2_	2359-15-1	1.32 × 10^4^	5.87 × 10^3^	1.09	2.26	1651
1-Guaiacyl-3-propanol	C_10_H_14_O_3_	2305-13-7	5.37 × 10^3^	2.11 × 10^3^	1.14	2.54	1651
Citronellyl tiglate	C_15_H_26_O_2_	24717-85-9	1.58 × 10^4^	7.03 × 10^3^	1.18	2.25	1658
1,8,11,14-Heptadecatetraene, (Z,Z,Z)	C_17_H_28_	10482-53-8	1.12 × 10^4^	5.03 × 10^3^	1.07	2.23	1664
3-Oxo-alpha-ionol	C_13_H_20_O_2_	34318-21-3	5.56 × 10^3^	2.31 × 10^3^	1.06	2.41	1647

**Table 2 insects-15-00810-t002:** Six candidate compounds purchased on the market.

Substances	Chemical Formula	Cas No.	Category
Bornyl acetate	C_12_H_20_O_2_	76-49-3	Terpenoids
α-Pinene	C_10_H_16_	80-56-8	Terpenoids
Cyclohexene	C_10_H_16_	586-62-9	Terpenoids
Methyl acrylate	C_6_H_10_O_2_	924-50-5	Ester
2-Oxepanone	C_6_H_10_O_2_	502-44-3	Ester
1-Nonanol	C_9_H_20_O	143-08-8	Alcohol

## Data Availability

The data presented in this study are available in the article.

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
