# Peer review of "Seed Coating with Thiamethoxam-Induced Plant Volatiles Mediates the Olfactory Behavior of *Sitobion miscanthi"

_insects, 2024, doi:10.3390/insects15100810_

Round 1
Reviewer 1 Report
Comments and Suggestions for Authors
The manuscript (MS) entitled “Seed coating with thiamethoxam-induced plant volatiles medi-2 ate the olfactory behavior of Sitobion miscanthi” by Sun et al., was aimed to investigate the impact of the pesticide Thiamethoxam (TMX) on the emission of Volatile Organic Compounds (VOCs) by wheat plants and how these VOCs are able to influence the behaviour of Sitobion miscanthi. In my opinion, the manuscript is very interesting, solid, and easy to follow. However, in my opinion some few points can be improved:
1. In the introduction section (lines 83-91) the aim of the study could be better presented;
2. In the Materials and Methods section, it is not clear how the VOCs emitted by wheat plants were collected, identified and quantified. In my opinion, it is necessary add a paragraph regarding these aspects.
3. In the Results section, about the analysis of chemical compounds, Authors reported the results of the Principal Component Analysis (PCA). However, in my view, it is necessary add a paragraph and a table to describe VOCs levels detected in wheat plants treated with TMX and untreated plants.
Author Response
Comments 1: In the introduction section (lines 83-91) the aim of the study could be better presented;
Response 1: Thank you for your valuable feedback. I have made the suggested modifications to the introduction section (now in lines 83-96), to better clarify the aims of the study. These changes aim to present the research objectives more clearly. I hope this revision addresses your concerns.
Comments 2: In the Materials and Methods section, it is not clear how the VOCs emitted by wheat plants were collected, identified and quantified. In my opinion, it is necessary add a paragraph regarding these aspects.
Response 2: Thank you for your valuable suggestions. We have made the necessary revisions as per your instructions, and these updates can now be found between lines 173 and 207. In this section, we have clarified the methodology and refined the analysis in accordance with your suggestions. We believe the revised version is clearer and addresses the concerns you raised. Please let us know if there are any further adjustments required.
Comments 3: In the Results section, about the analysis of chemical compounds, Authors reported the results of the Principal Component Analysis (PCA). However, in my view, it is necessary add a paragraph and a table to describe VOCs levels detected in wheat plants treated with TMX and untreated plants.
Response 3: Thank you for your feedback. We have made the necessary revisions as requested. The changes are now reflected in the updated table1, and in the lines 241 to 245 and 252 to 254. Please review these sections and let us know if any further adjustments are required.
Reviewer 2 Report
Comments and Suggestions for Authors
While I appreciate the effort the authors have made to develop their research, I believe they have had a lapse in submitting the paper to the publisher and the content does not match the abstract or, may be, few pages are missing.
I believe it is an interesting paper, and it represents a scientific contribution within the scope of this journal. Nevertheless, are many questions and corrections before the manuscript should be accepted for publication. I therefore recommend its publication with MAJOR corrections.
More important issues to answer
1. VOCs released by TMX-coated and non-coated wheat plants. It is necessary to introduce a table with the volatiles emitted by treated and untreated plants; quantification of each component, how its structure has been confirmed, retention times in GC, RI, etc. Also a statistical analysis of the components used in the bioassays.
2. Material and Methods. Chemicals, instrumentation, GC-MS procedure…….. Statistical analyses?????
3. 2.2. Aphid preference.......
I would advise changing the paragraph between lines 124 to 128, in front of the paragraph between lines 119 to 123.
4. Your literature review in the discussion section does not make sufficient use of the current scientific literature.
Minor comments
1. Figure 6 is difficult to understand, I recommend reworking it.
2. Abstract: Line 30 to 31. Please use quantities in mg, not concentrations (microliter/ml).
Author Response
More important issues to answer
Comments 1: VOCs released by TMX-coated and non-coated wheat plants. It is necessary to introduce a table with the volatiles emitted by treated and untreated plants; quantification of each component, how its structure has been confirmed, retention times in GC, RI, etc. Also, a statistical analysis of the components used in the bioassays.
Response 1: We have added a new table 1 as per your request, which includes essential information on the volatiles. This table provides details about the volatile compounds released. For statistical analysis of the components, we used the OPLS-DA model to calculate VIP values combined with fold changes to screen for differential substances (lines 241 to 245 and 252 to 254, Figure 3 and Figure 4). Please review this update and let us know if any further adjustments are required.
However, in terms of retention time, we have indeed identified metabolites by comparing the ion pairs, retention time, and fragmentation pattern in the database. As you say, retention time is very important for the annotation of each metabolite. However, due to the commercial conflict of interest, our cooperator (NoveBio Company) can’t provide the entire information of retention time. They are worried that their database may leach out and other commercial competitors may steal the company’s database once the information of retention time is published online. It means that the potential competitors would omit the step of database construction and earn profits rapidly if they obtain the database from NoveBio Company. In order to avoid commercial conflict of interest, we hope that you will also agree and appreciate this point.
For structural confirmation, mass spectrometry alone cannot to definitively determine the structure of a compound. Nuclear Magnetic Resonance (NMR) spectroscopy may be required for accurate structural identification. We kindly ask for your understanding regarding the inherent limitations and technical constraints involved in this process.
Comments 2: Material and Methods. Chemicals, instrumentation, GC-MS procedure…….. Statistical analyses?????
Response 2: Thank you for your feedback. We have made the requested revisions as per your instructions, and the modifications can be found in lines 173 to 207. In this section, we clarified the methodology and updated the analysis according to your suggestions. We hope the updated version will be clearer and address the issues you highlighted. Please let us know if further adjustments are needed.
Comments 3: 2.2. Aphid preference.......I would advise changing the paragraph between lines 124 to 128, in front of the paragraph between lines 119 to 123.
Response 3: We have made the changes according to your guidance.
Comments 4: Your literature review in the discussion section does not make sufficient use of the current scientific literature.
Response 4: Thank you for your valuable feedback. We have revised the discussion section to incorporate more recent and relevant scientific literature, particularly focusing on studies related to the olfactory behavior of aphids, the impact of pesticides on plant volatiles, and the approaches to pest management. The modifications can be found in lines 346 to 351, 357 to 363 and 379 to 390. We hope the revised version clarifies the points you mentioned and addresses the concerns raised. Should further revisions be necessary, please feel free to provide additional guidance.
Minor comments
Comments 1: Figure 6 is difficult to understand, I recommend reworking it.
Response 1: Thank you for your feedback regarding Figure 6(now Figure 8). To address your concern, we have included more detailed annotations between lines 332-337 to improve intelligibility. We hope this makes the figure easier to interpret and would appreciate your thoughts on the changes. Please feel free to let us know if further adjustments are necessary.
Comments 2: Abstract: Line 30 to 31. Please use quantities in mg, not concentrations (microliter/ml).
Response 1: Thank you for your suggestion regarding the use of quantities in milligrams. I understand the emphasis on using milligrams. However, the chemical standards I purchased are in liquid form, making it challenging to accurately convert to milligrams. Additionally, converting these standards to milligrams and then ensuring uniform concentration gradients across various metabolites could result in significant inaccuracies. I hope this clarifies the difficulty, and I appreciate your understanding.
Round 2
Reviewer 2 Report
Comments and Suggestions for Authors
Dear authors,
thank you very much for making some improvements to your original manuscript. In my humble opinion, for future similar studies, the collaboration of a natural products chemist or chemical ecologist would be of great help.